# Multidimensional Approach to Assess Nutrition and Lifestyle in Breastfeeding Women during the First Month of Lactation

**DOI:** 10.3390/nu13061766

**Published:** 2021-05-22

**Authors:** Andrea Gila-Díaz, Nuria Díaz-Rullo Alcántara, Gloria Herranz Carrillo, Pratibha Singh, Silvia M. Arribas, David Ramiro-Cortijo

**Affiliations:** 1Department of Physiology, Faculty of Medicine, Universidad Autónoma de Madrid, C/Arzobispo Morcillo 2, 28029 Madrid, Spain; andrea.gila@uam.es (A.G.-D.); nuriafitness@gmail.com (N.D.-R.A.); silvia.arribas@uam.es (S.M.A.); 2Division of Neonatology Hospital Clínico San Carlos, Instituto de Investigación Sanitaria del Hospital Clínico San Carlos (IdISSC), C/Profesor Martin Lagos s/n, 28040 Madrid, Spain; gherranz@gmail.com; 3Division of Gastroenterology, Beth Israel Deaconess Medical Center, Harvard Medical School, 330 Brookline avenue, Boston, MA 02215, USA; psingh6@bidmc.harvard.edu

**Keywords:** body composition, breastfeeding, diet intake, feeding behavior, food consumption, multidimensional scaling analysis

## Abstract

The first month of lactation is a vulnerable nutritional period for the mother. Our aims were (1) to evaluate the nutritional status of breastfeeding women in the first month of lactation, and (2) to explore different aspects of nutrition and lifestyle through a multidimensional approach. A total of 30 healthy breastfeeding women were enrolled in this study. Dietary pattern was assessed through a 72-hour dietary recall questionnaire (days 7 and 28 postpartum) and data were compared with Dietary Recommendation Values (DRV), and through the Adherence to Healthy Food Pyramid (HFP) questionnaire (day 28). Anthropometric parameters were evaluated by bioimpedance. Using factor analysis, nutritional dimensions were extracted, and linear regression models were used to analyze the association between anthropometric parameters and dimensions. Compared to the DRV, women showed insufficient energy, water, vitamin D, and potassium intake and excessive proteins, vitamins B1, B2, B3, B6, B12, and iron intake. We observed a moderate adherence to the HFP, with women being high in the fruits, vegetables, and oil categories, and low adherence to the physical activity, dairy products, and hydration categories. The nutritional dimension, including the HFP categories of physical activity, hydration, and animal protein intake together, was negatively associated with body weight (β = −3.7 ± 1.7; *p*-value = 0.047). In conclusion, during the first month postpartum, breastfeeding women exhibited several nutritional imbalances and poor physical activity negatively influencing anthropometric parameters. We propose a multidimensional approach to assess the nutritional status of breastfeeding women as a tool to detect specific deficiencies, allowing for personalized counseling.

## 1. Introduction

Maternal nutrition during pregnancy and lactation is critical for both mother and child’s health. During breastfeeding, nutritional and metabolic demands are considerably higher in lactating women [1], due to the energy expenditure involved in breastmilk (BM) production [2]. Lactation is a nutritionally vulnerable period for the woman, and if insufficient or inadequate, such as in low-income settings, it has an enormous impact on maternal health [3]. However, in high-income countries, where women have access to a sufficient amount of food, the nutrition of breastfeeding women may not be necessarily adequate, it may be in excess or unbalanced, as part of the general health problem of obesity in western countries. Moreover, during breastfeeding, the woman is primarily focused on the newborn and thus neglects her own diet. Maternal diet affects BM composition; in particular, levels of vitamin A, D, B1, B2, B3, B6, and B12, fatty acids, and iodine depend on maternal diet [4]. Together with diet, maternal body composition and fat stores also influence BM compounds. Therefore, a healthy diet ensures not only a good maternal nutritional status, but also optimal concentrations of important BM components and the newborn’s health. 

Despite the importance of maternal nutrition during lactation, in some countries, such as Spain, there are no specific nutritional recommendations for this period, and guidelines for anthropometric parameters are based on pregnant women or the non-breastfeeding adult population. This could be due to the lack of studies addressing women’s nutritional status during the lactation period compared to those during pregnancy. There is also an ongoing debate about supplementation during lactation; for example, how much and for how long the supplementation should be provided. The need for nutritional supplements depends on many factors, including socioeconomic status, cultural differences, and unhealthy maternal habits. In low-income settings, supplementation has been demonstrated as an effective strategy to compensate for nutritional deficiencies [5]. In high-income countries, supplementation has been mostly focused on long-chain fatty acids (LCPUFAs). Studies have shown that omega-3 (*n*-3) LCPUFA supplementation results in an increased level of docosahexaenoic acid in BM [6] and it is associated with improved neonatal health outcomes [7]. In European countries, efforts have been made—through an international research consortium—to summarize current evidence in order to provide guidelines and regulations on nutrition and supplementation in lactating women [8]. Despite important conclusions, mainly focused on the aspect of maternal nutrition to reduce the infant’s risk of overweight, this consortium has evidenced the need for additional studies to develop specific recommendations based on scientific evidence, which could lead to improved public health strategies [9,10,11]. Furthermore, an individualized approach to nutritional counseling considering women’s nutritional status and body composition has been proposed [12]. 

The Healthy Food Pyramid (HFP) is a simplified graphic representation of the main characteristics of the Mediterranean Diet. It is the main framework for the different sociocultural contexts of the Mediterranean region and aims to promote adherence to this healthy dietary pattern [13,14]. The benefits of this dietary pattern are well established [15,16]. A systematic review has confirmed that the Mediterranean Diet in lactating women decreases the risk of allergic disease and thus is beneficial for the newborn [17]. However, a gradual loss of this healthy food pattern was observed during pregnancy and the postpartum period [18]. In addition to the nutritional dimension, the HFP also takes into account other dimensions that include lifestyle or physical activity. Our research group has developed a questionnaire of adherence to the HFP (AP-Q) that analyzes these dimensions in addition to the nutritional condition, and thus provide a robust tool enabling one to assess several features that can affect overall maternal health status [19]. 

Breastfeeding is a vulnerable period for lactating women, particularly if the mother does not receive adequate nutritional recommendations. Our principal aim was to gain insight into the nutritional status of breastfeeding women in Spain during the first month of lactation. The objectives of the present study were to investigate whether the maternal nutritional pattern followed the dietary recommendations during the first month of lactation; their degree of adherence to the HFP, the dimensions that needed to be improved, and the association between nutritional and anthropometric variables. The knowledge gained by this study, using a multidimensional approach based on different dietary and lifestyle components, could help health professionals to design individualized interventions for breastfeeding women, based on the detected deficiencies.

## 2. Materials and Methods

### 2.1. Population of Study

Mothers were enrolled within the first 72 h postpartum at the Obstetrics and Gynecology and Neonatology Departments of the Hospital Clínico San Carlos (HCSC, Madrid; Spain). The enrollment period was from September 2019 to April 2020. Maternal inclusion criterion were: ≥18 years old, good understanding of Spanish language, and absence of diseases at the time of study. Mothers with dietary restrictions (i.e., diet for competitive sports or vegetarians/vegans), and death of newborns in the first 48 hours postpartum were excluded from the study. The mothers who agreed to participate in the study signed an informed consent. The final cohort included 30 women.

The present study was performed following the Declaration of Helsinki for studies on human subjects; it was approved by the Ethical Committee of HCSC from Madrid, Spain (Ref. 19/393-E, on June 2019).

### 2.2. Measurement of Maternal Dietary Patterns

Maternal dietary patterns were studied using two complementary surveys, (1) the 72-h dietary recall (72hDR) method and (2) the Adherence to Healthy Food Pyramid questionnaire (AP-Q).

**72-hour dietary recall (72hDR)**. This is an accurate and validated method to quantify an individual’s usual intake over a short period through open-ended questions [20,21,22]. The 72hDR was obtained at days 7 ± 1 and 28 ± 1 postpartum. The mothers indicated the ingredients, food preparation methods, and quantities of everything they ingested during the three previous days, including any supplements and water. Mothers were instructed using the “Visual Guide to Food and Rations” [23] to obtain detailed information. The different kinds of foods and their ingredients were recorded manually and classified according to the meals. Thereafter, the analysis was carried out using DIAL software (version 3.11.9, Alce Ingeniería, Madrid; Spain) to calculate nutrients intakes. The composition of nutritional supplements was also included in the software. The results were calculated for energy (kcal), water (mL), protein (g), fat (g), saturated fatty acids (SFAs; g), monounsaturated fatty acids (MUFAs; g), polyunsaturated fatty acids (PUFAs; g), total cholesterol (mg), carbohydrates (g), total dietary fiber (g), vitamins: A (retinol; μg), B1 (thiamin; mg), B2 (riboflavin; mg), B3 (niacin; mg), B5 (pantothenic acid; mg), B6 (pyridoxine; mg), B7 (biotin; μg), B9 (folic acid; μg), B12 (cobalamin; μg), C (ascorbic acid; mg), D (μg), E (α-tocopherol; mg), K (μg), and minerals including calcium (mg), iron (mg), iodine (mg), sodium (mg) and potassium (mg).

**Adherence to Healthy Food Pyramid questionnaire (AP-Q)**. This is a self-administered questionnaire developed and validated in the general population [19] that estimates the adherence to the HFP, which is a healthy standard for countries in the Mediterranean area [13,14]. It measures the frequency of consuming food of different categories during the last month. Mothers answered the AP-Q at 28 ± 1 days postpartum. The AP-Q consisted of 27 questions with multiple responses. The responses given to each item were grouped into ten different categories, including physical activity, healthy habits and culinary techniques, hydration, grains, seed and legumes, fruits, vegetables, oil type, dairy products, animal proteins, and snacks. Each category was scored on a scale of 0 to 1; the higher the score, the greater the adherence to the HFP. In HFP, the food categories at the bottom of the pyramid represent healthy habits (physical activity, culinary techniques, hydration, grains, seed and legumes, fruit and vegetables) and have positive scores, while categories at the top (snacks or foods that are not included in HFP) have negative scores.

### 2.3. Measurement of Maternal Anthropometric Parameters

Maternal anthropometric parameters were measured at days 7 ± 1 and 28 ± 1 postpartum. Height (cm) was quantified by stadiometer (Seca 217, TAQ sistemas medicos, Madrid, Spain), and waist and hip circumferences (cm) were measured using anthropometric tape with millimeter precision (Seca 201, TAQ sistemas medicos). The waist-to-hip ratio was also calculated. Body weight (kg), total fat and muscle mass (%), and basal metabolic rate (kcal/day) were estimated using a bioimpedance meter (Omron Healthcare HBF-514C Full Body Sensor W Scale, Madrid, Spain), according to the manufacturer’s instructions. The body mass index (BMI; kg/m^2^) was also calculated.

### 2.4. Statistical Analysis

The Shapiro–Wilk test was used to examine the distribution of the variables and quantitative measurements were either expressed as mean ± standard error of the mean (SEM), for those with normal distribution, or as a median and interquartile range (Q1; Q3) for variables that did not follow a normal distribution. Qualitative variables were reported as a percentage (%) along with the sample size. To compare the maternal diet with the given dietary reference value (DRV), one-sample Student’s *t*-test or Wilcoxon signed-rank test were performed. The statistical difference in diet between day 7 and 28 of lactation was analyzed by paired Student’s *t*-test or Wilcoxon signed-rank test with continuity correction. The AP-Q categories between women with term and preterm delivery were analyzed using the Student’s *t*-test.

Th sphericity of the dietary variables was evaluated by Barlett’s test and Kaiser–Meyer–Oklin (KMO) score. For cluster analysis, KMO score ≥0.7 and a significant Barlett’s test were considered adequate. Thereafter, the dimensions with eigenvalues ≥1 were extracted, according to Kaiser’s rule. The grouping was carried out according to dimensions. The weight of each dietary dimension was calculated by the standardized loading of dietary variables within a given dimension. The standardized loading was extracted from the varimax-rotated matrix, considering a cut-off ≥0.45 to avoid overlapping in the dietary variables. Correlations were performed by the Rho-Spearman test and presented as a correlation matrix. 

Linear regression models were used to study the association between anthropometric parameters and dietary dimensions after adjusting for maternal age (continuous), gestational age (continuous), ethnicity (categorical), educational level (categorical), employment status (categorical), and supplementation (yes/no) during the lactation period. Estimated beta coefficient ± standard error was reported. 

All data analyses were performed using R software (version 3.5.2, R Core Team 2018) within RStudio (version 1.1.453, RStudio, Inc., Vienna, Austria) using the tidyverse, compareGroups, cluster, factoextra, and fcp packages. Plots, clusters, dendrogram and correlogram were generated using ggplot2, ggdendro, ggpubr and ggcorrplot packages. Results were considered statistically significant for *p*-value ≤ 0.05.

## 3. Results

The mean maternal age was 34.5 ± 1.0 years (max = 44.0; min = 22.0). The majority of the women were of Caucasian origin (78.6% (22/28)), mainly Spanish (71.4% (20/28)). The rest of the mothers were of Latin-American (14.3% (4/28)) or Asian (7.1% (2/28)) origin. Regarding educational level, 46.4% (13/20) had university degrees, and 53.6% (15/28) had high school degrees. In our cohort, 75.0% (21/28) of the mothers were employed. The average gestational age was 34.8 ± 0.9 weeks (max = 41.4; min = 27.1). For 36.7% (11/30) of the mothers, this was the first child; 24.0% of the population was diagnosed with vitamin D deficiency during pregnancy, and 57.1% (16/28) of the women were taking breastfeeding supplementation. Regarding breastfeeding supplementation, 62.5% (10/16) of the women took *Natalbén-Lactancia*^®^ [24]. This supplement was taken at a dose of two pills/day; details of the composition are shown in Appendix A. 

### 3.1. Maternal Dietary Pattern during the First Month of Lactation 

Maternal diet composition was assessed at days 7 and 28 postpartum. At day 7, the percentage of macronutrients in the diet was: protein = 18.3 ± 0.6%, fat = 40.7 ± 1.4% and carbohydrates = 41.0 ± 1.5% and at day 28: protein = 19.2 ± 0.5%, fat=41.5 ± 1.0% and carbohydrates = 39.3 ± 1.0%. 

We did not detect statistical differences in any of the macro or micronutrients between day 7 and 28 postpartum (Table 1; *p*-value ^2^). 

To prove whether the maternal diet of the cohort was in accordance with recommendations during the first month of breastfeeding [12,25,26], we compared and analyzed maternal nutrient intake at day 7 and 28 with the recommended values. We found that maternal diet was significantly deficient in energy, water, vitamin D, and potassium at day 7 and 28 compared with the DRV. In contrast, the levels of protein, vitamins B1, B2, B3, B6, B12, and iron were significantly higher at both time points, compared to the DRV. Saturated fatty acids intake was high on days 7 and 28 since the recommended intake is “as low as possible” (Table 1). The levels of vitamin A and B9 were significantly lower on day 7, while the levels of vitamin B7 and E were significantly higher on day 28 compared to the recommendations (Table 1; *p*-value ^1^).

In the cohort, the AP-Q categories scored as follows: physical activity = 0.25 ± 0.05; healthy habits and culinary techniques = 0.55 ± 0.04, hydration = 0.51 ± 0.06, grains, seed and legumes = 0.56 ± 0.05, fruits = 0.85 ± 0.05, vegetables = 0.71 ± 0.04, oil type = 0.64 ± 0.08, dairy products = 0.46 ± 0.01, animal proteins = 0.54 ± 0.03 and snacks = 0.64 ± 0.03. Our data indicate that mothers scored high in fruits, followed by vegetables, and oil type categories, and scored low in physical activity, dairy products, and hydration. The data are also represented divided by quartiles in Figure 1. No differences were detected between mothers with preterm and term delivery (Appendix A).

### 3.2. Factor Analysis for Maternal Dietary Pattern

Cluster analysis was performed using all dietary variables on day 28. The suitability of this analysis was demonstrated by the significant sphericity (χ^2^ = 2699.2; *p*-value < 0.001) and a KMO score of 0.93. Thereafter, Kaiser’s rule was applied to extract the optimal number of dimensions. We found six dimensions with eigenvalues ≥ 1, which explained 67% of the variance. Dimension 1–6 accounted for 17.2%, 14.8%, 11.7%, 9.6%, 7.7%, and 5.9% of explained variance, respectively. The macronutrients were distributed in dimension 1. In dimension 2, most of the vitamins and the healthy habits category were found. In dimension 3, fiber, vitamin K, potassium, grain, and snacks categories were clustered. Dimension 4 included fruit, vegetables, dairy products, and oil categories. Dimension 5 included iodine and iron, and in dimension 6 physical activity, animal protein, and hydration categories clustered together (Appendix A).

### 3.3. Maternal Anthropometric Parameters and Relationship with Maternal Dietary Pattern

Changes in maternal anthropometric parameters from day 7 to day 28 are shown in Table 2. From day 7 to day 28 of lactation, a significant decrease in maternal body weight, BMI, waist circumference, waist-to-hip ratio, muscle mass, and basal metabolic rate was detected. In contrast, a significant increase in fat mass was observed during this period.

The correlations between anthropometric parameters and the dietary pattern dimensions are presented as a correlation matrix in Figure 2. As expected, a positive correlation was observed between the anthropometric variables, as well as between the different dimensions. 

Regarding the relationship between anthropometric parameters and dietary pattern dimensions, dimensions 1 and 6 showed significant correlations. Dimension 1 (energy, water intake, fats, SFAs, MUFAs, PUFAs, cholesterol, carbohydrates, proteins, sodium, and calcium) was positively and significantly correlated with muscle mass (rho = 0.36; *p*-value = 0.009), and it was negatively correlated with waist circumference (rho = −0.37; *p*-value = 0.008). Dimension 6 (physical activity, hydration, and animal protein of the AP-Q categories) was significantly and negatively correlated with body mass index (rho = −0.48; *p*-value = 0.002), fat mass (rho = −0.43; *p*-value = 0.004), weight (rho = −0.46; *p*-value = 0.002), and waist circumference (rho = −0.38; *p*-value = 0.006).

To assess whether there was an association between the anthropometric parameters and dimensions 1 and 6, linear regression models, adjusted for maternal age, gestational age, ethnic group, educational level, employment situation, and breastfeeding supplementation, were built (Table 3). We did not find a significant association between dimension 1 and any of the anthropometric parameters. Dimension 6 was negatively and significantly associated with body weight and was close to the significance level for waist circumference. 

## 4. Discussion

In this study, we evaluated the nutritional status of breastfeeding women from different perspectives in order to explore different areas that may be susceptible to intervention. Here, we showed that the diet of the breastfeeding women in our cohort, with a middle–upper socioeconomic status in Spain, is imbalanced in several nutrients. Moreover, women receive supplementation on a regular basis, which may not always be needed. Their dietary pattern showed a moderate adherence to the HFP, with the worse aspects being physical activity, hydration and dairy products. We also found that dimension 6 (which includes physical activity, hydration, and animal proteins) was negatively associated with maternal body weight and waist circumference, suggesting the impact of dimension 6 on maternal anthropometric parameters during the first month of lactation. We conclude that the proposed multidimensional approach evaluating different aspects of nutrition and lifestyle, could help to identify specific deficiencies in breastfeeding women, allowing for personalized counseling during this important period of life. Our data also evidence the importance of addressing the normal range for maternal anthropometric parameters during the first month of lactation in larger cohorts. These data will be valuable for personalized counseling during this important period of life.

### 4.1. Breastfeeding Women’s Diet Show Nutritional Differences for Dietary Recommendation Values

Our results showed that during the first month postpartum, breastfeeding women were deficient in energy, water intake, vitamin D, and potassium compared with the recommendations for the breastfeeding period. On the other hand, the intake of proteins, saturated fatty acids, vitamin B complex and iron were above the recommended levels. For calorie consumption, our study population was below the recommendations, specifically on day 28 (1870 kcal/day). It has been demonstrated that calorie intake less than 1800 kcal/day results in decreased milk production and macronutrient concentration in BM, which may be clinically relevant [27]. 

Several studies have shown that the Spanish population tends to reduce energy intake and the quality of the diet is gradually deteriorating, moving away from the Mediterranean Diet. Our data showed that the intake of protein and saturated fatty acids exceeded recommendations, which could lead to a decreased carbohydrate consumption, as previously described in adult population [28,29]. A high intake of saturated fatty acids might result in increased cholesterol levels that might have negative consequences for maternal health [30]. However, we found an adequate intake of polyunsaturated fatty acids. We did not detect whether there was an adequate consumption of the different n-3 and n-6 LCPUFAs, since DRV only includes linoleic, linolenic acids, and the combination of eicosapentaenoic and docosahexaenoic acids. It would be important to have reference values for all n-3 and n-6 LCPUFAs to achieve an adequate intake. Knowledge of the DRV of these important fatty acids during the breastfeeding period could be used for counseling, since the Western diet is usually misbalanced, with an increased n-6 and decreased n-3 intake, which plays an important role in obesity [31]. Besides, maternal supplementation with n-3 LCPUFAs results in increased levels of docosahexaenoic acid in BM [6], which is associated with improved neonatal health outcomes [7].

We found a low water intake in our study population, which could be due to failure to recall it in the 72hDR questionnaire. However, the data of the AP-Q-hydration category confirmed the low water intake, suggesting that the importance of hydration was neglected in breastfeeding women and needs to be improved through counseling. Besides, the similarity between the information provided by both questionnaires supports the utility of the AP-Q. Our data are in accordance with a previous report, showing insufficient total water intake in breastfeeding women, which also evidenced that water in food was the principal source [32]. There is no clear evidence showing the impact of the amount of water intake in breastfeeding women on BM production [33]. A lack of well-conducted trials and the impact of hydration on BM production deserves further attention [34].

Regarding vitamin intake, our data demonstrated that the consumption of water-soluble vitamins is above the recommended levels. This may be related to the fact that the vast majority of the mothers had a high intake of fruits and vegetables, with good adherence to the HFP, and they were also taking supplements. Since there is a relationship between maternal consumption of water-soluble vitamins and their content in BM [35], it is possible that the level of these vitamins may be adequate for BM. On the other hand, the vitamin A, vitamin D, and folic acid, were below the recommended levels. Vitamin A was low at day 7 but improved by day 28, suggesting that they probably had a diet deficient in vitamin A during gestation (at least by the end of gestation). In western countries, vitamin A supplements are not suggested for pregnant women who consume the recommended dietary allowance during their reproductive years. Studies have suggested that increased consumption of vitamin A-rich foods can meet increased demands during the lactation period [36]. Our data suggest that it would be important to detect women with diets deficient in Vitamin A to provide an appropriate recommendation. Vitamin D in our population was far below the recommended values. Besides, 24% of the cohort had a vitamin D deficiency clinically determined in pregnancy, which could be related to an inadequate consumption of food containing Vitamin D as well as a low sun exposure [37]. Dairy products provide a natural food source for Vitamin D, in many cases such as cow’s milk, because they are often fortified with this vitamin. The 72hDR data showed that vitamin D intake is in accordance with previous reports, suggesting significantly decreased Vitamin D in recent decades [38]. Vitamin D is transferred from maternal serum to BM [39,40] suggesting that infants of mothers who consume less Vitamin D could also be deficient. Therefore, dietary counseling is clearly needed in breastfeeding women. During pregnancy, supplementation is usually provided to women with deficiency. Our study indicates that, if the diet is poor in Vitamin D, it may be beneficial to continue supplementation during the breastfeeding period. This would benefit both the mother and the infant since it has been demonstrated that maternal supplementation, as well as exposure to the sun, increases vitamin D in breastmilk [41]. 

Our data also showed insufficient levels of folic acid in maternal diet at the beginning of the breastfeeding period, but at day 28 these levels were within the recommended values. These results could be possibly due to the discontinuation of prenatal folic acid supplements after childbirth [42] followed by a diet improvement afterward during the lactation period. However, given our results, it may be useful to recommend maintaining folic acid supplementation during the first month of lactation if the maternal diet does not provide sufficient amounts. Substantial variations in folate recommendations for lactating women are found across countries (260–650 µg/day) [43]. These measurements are based on BM folate levels. We suggest that it would be more appropriate to detect maternal diet deficiencies through nutritional questionnaires, in order to give nutritional counseling or provide supplementation to women if needed. 

In our study population, vitamin E levels were high on day 28. Vitamin E is particularly important at the early stages of life, as it acts as a defense against oxygen-induced toxicity in the extrauterine environment. BM is responsible for supplying the demand of vitamin E to the neonates [44]. Vitamin E is present in vegetables, oils, nuts, and seeds, and the adequate intake in our population is likely related to the consumption of these food categories, evidenced through the AP-Q.

Regarding minerals, calcium, iodine, and sodium levels were adjusted to the recommendations throughout the first month of lactation. Although daily calcium intake is well below the recommended levels in many countries, particularly in low-income settings [45], our cohort showed adequate levels, which may be related to the supplementation. Calcium supplementation during breastfeeding has some controversies. On one hand, it has been proposed that it may damage iron status, although this has been refuted by studies demonstrating that long-term calcium supplementation did not impair iron stores in breastfeeding women [45,46]. On the other hand, there is a paradox with calcium supplementation in pregnant women with a calcium-poor diet, and it has been reported that it may harm the woman’s skeletal system [47]. We suggest that it would be wise to have specific information on the nutritional status of each mother to detect those who can benefit from supplementation.

Potassium intake was below the recommended values. Potassium levels may influence the physiological parameters of the mother, such as blood pressure. In fact, a diet rich in potassium has been shown to have blood pressure-lowering effects [48]. The major sources of potassium are lentils, red beans, nuts, dairy products, meat, poultry, and fish [49]. Considering the important role of potassium and its balance with sodium, increased intake of dairy products and cereals would be beneficial for breastfeeding women. 

In our study population, iron intake exceeds (up to 90 mg) the recommendations. This exceeding level of iron could be due to the iron supplementation during breastfeeding. However, excessive iron intake may result in increased oxidative stress and impaired glucose metabolism [50], suggesting that it would be important to reduce these doses or even eliminate them if there is no iron deficiency.

### 4.2. Anthropometric Parameters

During pregnancy, the maternal body undergoes several metabolic adaptations, such as increased fat mass, as a reserve of energy stores to supply the increased energy demand for the developing fetus, and for mammary gland growth to ensure and maintain lactation [1,51]. Therefore, pregnancy and postpartum are periods of increased vulnerability for the retention of excess body fat. In industrialized societies, being overweight in the perinatal period is a relevant problem and has a negative impact on maternal and neonatal health. Overweight and obese women initiate breastfeeding later and have a shorter duration [52], and obesity negatively affects BM composition [53].There is evidence that lactation promotes maternal weight loss through the mobilization of the fat deposits accumulated during pregnancy [54]. It has been shown that women who breastfed for more than 6 months had less central adiposity and reduced body weight retention [55] later on compared to those who did not breastfeed [56]. A systematic review showed that this effect is markedly dependent on the time points measured and breastfeeding intensity, being mostly observed with prolonged periods of lactation [54]. Few studies have focused on the body composition changes during the first month of lactation to establish the average anthropometric values during early postpartum. Our data show that during the first month, women reduce their body weight and BMI. However, according to the bioimpedance data, weight loss was related to lean mass reduction, while fat mass was increased. Our data are in accordance with a recent study in Mexico, showing reduced BMI together with a lean mass between the 8–16th week postpartum [57]. Although BMI has been taken as the gold standard for fat mass, increasing evidence indicates that it may not always be a good parameter to predict obesity, at least in some populations [58,59,60]. In accordance with these reports, our data also suggest that BMI might not be a good indicator of overweight or obesity in breastfeeding women, at least in the early stages of lactation. These results suggest that other tools are needed to distinguish cases with normal BMI but with high-fat mass, such as waist-to-hip ratio or fat mass measured by other methods such as bioimpedance or DEXA. To the best of our knowledge, there are no specific anthropometric reference values for breastfeeding women at different states of lactation. We suggest that it would be important to obtain such values through large cohort studies.

We also evaluated the relationship between anthropometric parameters and dietary pattern dimensions, which could help to provide specific counseling to women to improve their anthropometry and health outcomes. Our cluster analysis was performed using all dietary variables on day 28. It resulted in six dimensions, which explained the 67% of the variance. Dimensions 1 and 6 were significantly correlated with the anthropometric parameters. Furthermore, dimension 6 (physical activity, hydration, and animal protein) exhibited a negative association with body weight and waist circumference. These data suggest that improved physical activity, hydration, and high intake of high-quality animal proteins could have a beneficial impact on maternal anthropometry, such as reducing body weight and waist circumference. The relevance of physical activity was also evidenced by the AP-Q, which showed a low adherence in our population. This is likely related to the fact that in the first month postpartum, the mother needs to recover from delivery, and she is more focused on the infant’s health than her own. However, although we did not detect differences in the AP-Q between mothers with and without preterm delivery, it should be noted that mothers with premature infants, who are usually hospitalized during the first month of life, could skew feeding and adherence to the HFP. Around 60% of the population studied had other children, which could modulate their dietary and life-style pattern. This aspect was not explored in the present study, but deserves further analysis, comparing women who have their first child and those with previous lactation experience. Physical inactivity during the breastfeeding stage would likely lead to a loss of lean mass. This could also be related to insufficient energy consumption, as shown in our cohort. Although we detected an excessive consumption of proteins during the first month, this consumption may be of low quality. Furthermore, they can have excess saturated fats, which may result in an increased fat mass during the first month of lactation. Our data suggest that it would be important to provide specific recommendations to the mothers on the consumption of high-quality proteins, such as those present in oily fish, which also would improve the ratio n-3:n-6.

## 5. Conclusions

This work focuses on breastfeeding women, a population that may be nutritionally vulnerable, particularly during the first month of postpartum. Our main conclusions are:During the first month of lactation, the mothers did not satisfy the nutritional requirements of the guidelines. The important deviations in some micronutrients suggest that the women would benefit more from individualized supplementation, based on their nutritional status.Adherence to the HFP was moderate, and increased physical activity and dairy product consumption could be beneficial for maternal health.BMI may not be a good indicator of obesity in breastfeeding women during early postpartum. Larger cohort studies would help to determine the normal body composition values in this population.The novel multidimensional approach used in this study enables us to evaluate possible deficiencies in specific dietary components and health habits, which could enable personalized interventions in those areas that require attention.Studies in larger cohorts would be important to provide specific recommendations adapted to the healthy food pattern of different populations.

## Figures and Tables

**Figure 1 nutrients-13-01766-f001:**
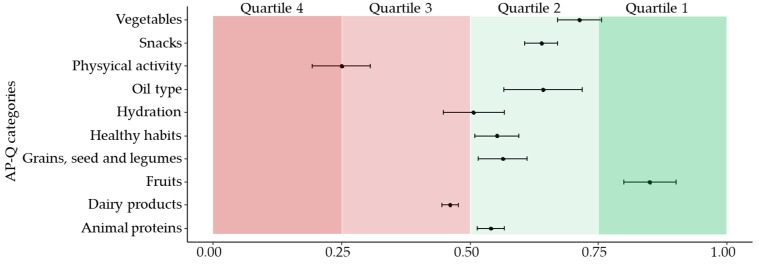
Scores of the AP-Q categories, segmented by quartiles. Data show the median ± standard error of the mean. AP-Q categories range from 0 to 1, being 1 high adherence HFP guidelines.

**Figure 2 nutrients-13-01766-f002:**
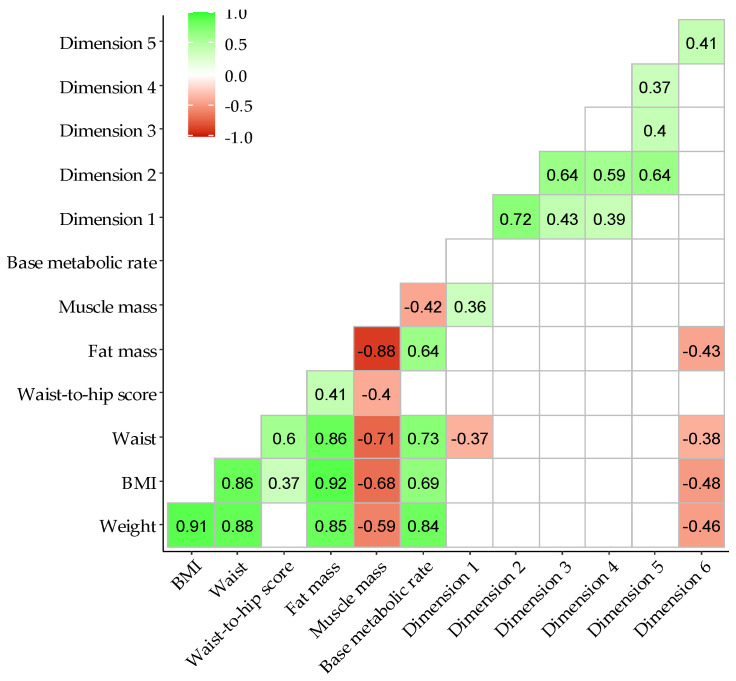
Correlogram between maternal body composition and dietary pattern dimensions ordered by hierarchical cluster. Body mass index (BMI). The legend represents Spearman’s rho coefficient with *p*-value ≤ 0.05. White grid were coefficients with *p*-value > 0.05.

**Table 1 nutrients-13-01766-t001:** Maternal nutrients consumption on days 7 and 28 postpartum and comparison with recommended values.

	DRV	Day 7 (*n* = 30)	*p*-Value ^1^	Day 28 (*n* = 22)	*p*-Value ^1^	*p*-Value ^2^
Energy (kcal)	2800	↓2101 (1754; 2437)	<0.001	↓1870 (1694; 2248)	<0.001	0.51 ^a^
Water (mL)	2700	↓1653 (1317; 2382)	<0.001	↓1894 (1261; 2868)	0.010	0.32 ^a^
**Macronutrients**						
Protein (g)	66	↑93.4 ± 26.7	<0.001	↑93.1 ± 22.0	<0.001	0.59 ^b^
Fat (g)	-	92.9 ± 28.6	-	91.5 ± 28.7	-	0.26 ^b^
Saturated fatty acids (g)	ALAP	↑31.9 ± 11.0	-	↑29.9 ± 10.3	-	0.18 ^b^
Monounsaturated fatty acids (g)	-	39.9 ± 12.5	-	39.2 ± 14.4	-	0.26 ^b^
Polyunsaturated fatty acids (g)	-	12.6 (8.9; 15.1)	-	12.7 (11.2; 15.8)	-	0.74 ^a^
Cholesterol (mg)	-	385.0 ± 142.0	-	375.0 ± 139.0	-	0.35 ^b^
Carbohydrates (g)	210	198.0 (158.0; 268.0)	0.88	180.0 (149.0; 209.0)	0.06	0.84 ^a^
Dietary fiber (g)	-	19.9 (13.2; 27.4)	-	18.8 (15.3; 27.8)	-	0.67 ^a^
**Vitamins**						
Vitamin A (Retinol; μg)	1300	↓1046 (756; 1340)	0.022	1381 (718; 1947)	0.61	0.23 ^a^
Vitamin B1 (Thiamin; mg)	1.4	↑1.8 (1.3; 2.2)	0.001	↑2.3 (1.4; 2.5)	0.004	0.68 ^a^
Vitamin B2 (Riboflavin; mg)	1.7	↑2.2 (1.7; 2.6)	0.001	↑3.0 (2.0; 3.4)	0.001	0.21 ^a^
Vitamin B3 (Niacin; mg)	17	↑43.8 ± 14.7	<0.001	↑46.3 ± 11.5	<0.001	0.61 ^b^
Vitamin B5 (Pantothenic acid; mg)	7.0	5.9 (5.4; 10.2)	0.74	↑10.2 (5.4; 12.3)	0.015	0.61 ^a^
Vitamin B6 (Pyridoxine; mg)	1.7	↑2.9 ± 1.2	<0.001	↑3.2 ± 1.3	<0.001	0.83 ^b^
Vitamin B7 (Biotin; μg)	45	38.3 (26.4; 75.4)	0.80	↑64.7 (29.4; 82.2)	0.022	0.99 ^a^
Vitamin B9 (Folic acid; μg)	500	↓306 (248; 433)	0.001	498 (372; 602)	0.73	0.12 ^a^
Vitamin B12 (Cobalamin; μg)	5	↑6.8 ± 3.2	0.003	↑7.0 ± 2.6	0.001	0.55 ^b^
Vitamin C (Ascorbic acid; mg)	155	142.0 (74.7; 194.0)	0.45	155.0 (129.0; 273.0)	0.29	0.47 ^a^
Vitamin D (μg)	15	↓3.4 (1.4; 6.5)	<0.001	↓4.8 (2.6; 8.0)	<0.001	0.61 ^a^
Vitamin E (α-Tocopherol; mg)	11	10.4 (7.0; 17.3)	0.63	↑17.9 (9.3; 20.1)	0.007	0.32 ^a^
Vitamin K (μg)	90	106.0 (64.4; 161.0)	0.06	94.4 (68.3; 181)	0.10	0.67 ^a^
**Minerals**						
Calcium (mg)	950	986 (748; 1155)	0.89	1094 (762; 1226)	0.24	0.54 ^a^
Iron (mg)	16	↑18.1 (13.2; 29.4)	0.038	↑27.6 (18.8; 92.2)	<0.001	0.14 ^a^
Iodine (mg)	200	103 (75; 284)	0.11	268 (109; 300)	0.30	0.42 ^a^
Sodium (mg)	2000	2123 (1543; 2461)	0.78	1852 (1724; 2262)	0.80	0.18 ^a^
Potassium (mg)	4000	↓3051 ± 788	<0.001	↓3271 ± 1047	0.004	0.49 ^b^

Data show median and interquartile range (Q1; Q3) or mean±standard error of the mean, according to the distribution of the variable. Dietary Recommendation Values (DRV) according to the population reference intake of a nutrient that is likely to meet the needs of a healthy population or the average requirement refers to the intake of a nutrient that meets the daily needs of people in a healthy population [12,25,26] for breastfeeding population; as low as possible (ALAP). ^1^ Comparison with DRV (↓ = below DRV; ↑ = above DRV); ^2^ comparison between day 7 and 28 (^a^ Wilcoxon signed-rank test with continuity correction; ^b^ paired Student’s t-test). Arrows indicate deviations from DRV.

**Table 2 nutrients-13-01766-t002:** Comparison of maternal anthropometric parameters from day 7 to day 28 of lactation.

	Day 7 (*n* = 28)	Day 28 (*n* = 28)	*p*-Value
Body weight (kg)	69.8 (63.3; 79.5)	68.9 (63.1; 78.4)	0.006 ^a^
Height (cm)	162.0 (160.0; 164.0)	162.0 (160.0; 164.0)	0.42 ^a^
Body mass index (kg/m^2^)	26.5 (24.2; 30.3)	25.7 (24.2; 29.5)	0.007 ^a^
Waist circumference (cm)	94.0 (85.0; 103.0)	92.0 (85.8; 100.0)	0.019 ^a^
Hip circumference (cm)	105.0 (102.0; 113.0)	106.0 (102.0; 113.0)	0.93 ^a^
Waist-to-hip ratio	0.90 ± 0.06	0.88 ± 0.07	0.029 ^b^
Fat mass (%)	39.2 ± 7.2	39.8 ± 7.3	0.030 ^b^
Muscle mass (%)	26.1 (25.0; 27.4)	25.4 (24.3; 26.8)	0.001 ^a^
Basal metabolic rate (kcal/day)	1400 (1322; 1504)	1374 (1305; 1457)	0.014 ^b^

Data show median and interquartile range (Q1; Q3) or mean±standard error or mean, according to the distribution of the variable. ^a^ Wilcoxon signed-rank test with continuity correction; ^b^ paired Student’s t-test.

**Table 3 nutrients-13-01766-t003:** Linear regression models to study the association between maternal anthropometric parameters and dietary pattern correlated dimensions.

	Dimension 1		Dimension 6	
	Estimated β ± SE	*p*-value	Estimated β ± SE	*p*-value
Body weight	-	-	−3.7 ± 1.7	0.047
Body mass index	-	-	−1.1 ± 0.6	0.09
Waist circumference	−0.3 ± 0.3	0.37	−2.9 ± 1.4	0.05
Fat mass	-	-	−1.3 ± 0.7	0.11
Muscle mass	0.06 ± 0.05	0.35	-	-

Data show estimated beta coefficient ± standard error (SE) and its associated *p*-value. The models were adjusted by maternal age, gestational age, ethnicity, study level, employment status, and breastfeeding supplementation intake.

## Data Availability

The data presented in this study are available on request from the corresponding author. The availability of the data is restricted to investigators based in academic institutions.

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
