# Peer review of "Multidimensional Approach to Assess Nutrition and Lifestyle in Breastfeeding Women during the First Month of Lactation"

_nutrients, 2021, doi:10.3390/nu13061766_

Round 1
Reviewer 1 Report
The authors focused on the first month of lactation as a period of nutritional vulnerability for the mother with the objective of assessing the nutritional status of breastfeeding women during the first month of lactation, and exploring different aspects of nutrition and lifestyle through a multidimensional approach.
This study includes 30 healthy lactating women.
Using a multidimensional approach, the authors assess the nutritional status of lactating women and detect specific deficiencies that influence anthropometric parameters. The results of this study will allow personalized counseling.
Author Response
Response: Thank you for your time to review this manuscript.
Reviewer 2 Report
The work is very interesting, it shows new data on nutritional behavior in 1 month of breastfeeding. The combination of anthropometric and nutritional data adds value to the work presented. I have a few doubts and questions for the authors:
line 182 - The authors give the average age of the examined mothers: 34.5 years. Quite high age, minimal and max data are missing, has the influence of maternal age on feeding method been studied?
did the authors take into account that the respondents had a child before. It is very often a differentiating variable - women who had a baby earlier and those in the 1st pregnancy.
line 491 - no literature is given in the conclusion section. These are the results of the researchers. Please remove the citation or the entire application.
Author Response
The work is very interesting, it shows new data on nutritional behavior in 1 month of breastfeeding. The combination of anthropometric and nutritional data adds value to the work presented.
Response: Thank you for reviewing our manuscript and your suggestions. Below are our responses to your suggestions. We have added in the title “during the first month of lactation”.
I have a few doubts and questions for the authors:
- Line 182 - The authors give the average age of the examined mothers: 34.5 years. Quite high age, minimal and max data are missing, has the influence of maternal age on feeding method been studied?
Response: We agree with the reviewer’s comment. The age of the women in our study reflects the present sociodemographic situation in the Spanish population, with a gradual increase in childbearing age. This fact has also been described in other European countries (https://ec.europa.eu/eurostat/web/products-eurostat-news/-/DDN-20210224-1). For this reason, our linear regression models were adjusted by maternal age, among other variables. In addition, following the reviewer’s comment, we have added maximum and minimal age (line 181).
- Did the authors take into account that the respondents had a child before. It is very often a differentiating variable - women who had a baby earlier and those in the 1st pregnancy.
Response: This is a very interesting insight. Although we have not explored this aspect in this manuscript, we have included it in the discussion (line 468-471). Furthermore, we have updated the results with the percentage of women in the study who had their first child (line 186).
- Line 491 - no literature is given in the conclusion section. These are the results of the researchers. Please remove the citation or the entire application.
Response: According to the reviwer’s comment, the reference [61] in the conclusion section was removed.